# Uncovering the functional anatomy of the human insula during speech

Oscar Woolnough[1,2], Kiefer James Forseth[1,2], Patrick Sarahan Rollo[1,2], Nitin Tandon[1,2,3]*

[1]Vivian L. Smith Department of Neurosurgery, McGovern Medical School at UT Health Houston, Houston, United States; [2]Texas Institute for Restorative Neurotechnologies, University of Texas Health Science Center at Houston, Houston, United States; [3]Memorial Hermann Hospital, Texas Medical Center, Houston, United States

**Abstract** The contribution of insular cortex to speech production remains unclear and controversial given diverse findings from functional neuroimaging and lesional data. To create a precise spatiotemporal map of insular activity, we performed a series of experiments: single-word articulations of varying complexity, non-speech orofacial movements and speech listening, in a cohort of 27 patients implanted with penetrating intracranial electrodes. The posterior insula was robustly active bilaterally, but after the onset of articulation, during listening to speech and during production of non-speech mouth movements. Preceding articulation there was very sparse activity, localized primarily to the frontal operculum rather than the insula. Posterior insular was active coincident with superior temporal gyrus but was more active for self-generated speech than external speech, the opposite of the superior temporal gyrus. These findings support the conclusion that the insula does not serve pre-articulatory preparatory roles.

*For correspondence:
Nitin.Tandon@uth.tmc.edu

Competing interests: The authors declare that no competing interests exist.

## Introduction

Multiple lesion studies have linked insular damage to speech and orofacial motor control disorders such as apraxia of speech (AOS) (*Dronkers, 1996*; *Marien et al., 2001*; *Ogar et al., 2006*; *Itabashi et al., 2016*), dysarthria (*Baier et al., 2011*) and dysphagia (*Daniels et al., 1996*). Dronkers' study was the first to quantitatively link anterior insular damage to a disruption in speech production, finding a 100% lesion overlap in the left superior precentral gyrus of the insula (SPG) in AOS patients. Lesion symptom mapping revealed that patients with SPG lesions produced a greater number of speech errors during complex, multisyllabic articulations (*Ogar et al., 2006*; *Baldo et al., 2011*). This implicated the insula in pre-articulatory motor plans specific to speech. Further evidence was contributed by fMRI studies that implicate the anterior insula in both speech perception and production (*Mutschler et al., 2009*; *Adank, 2012*; *McGettigan et al., 2013*; *Ardila et al., 2014*; *Oh et al., 2014*). In the dual-stream model of the speech articulation network, the anterior insula is included as part of the putative dorsal sensorimotor pathway (*Hickok and Poeppel, 2007*).

However, the data regarding insula's direct involvement in speech are not all concordant. Comparative assessments of activation to speech and non-speech oral movements have shown greater activity for non-speech within the insula (*Bonilha et al., 2006*) suggesting a more general role in oral motor control without specialization for speech. Also, it has been suggested that the apparent involvement of the SPG in complex articulation might actually be attributable to the inferior frontal gyrus (IFG) (*Fedorenko et al., 2015*), therefore the insula's engagement based on AOS lesion studies may simply reflect the high probability of insular damage following middle cerebral artery ischemia, with Broca's area lesions being more causative of AOS (*Hillis et al., 2004*).

Experiments directly stimulating the insula have also been inconclusive. Of the many patients who have undergone insular stimulation with implanted electrodes, somatosensory manifestations are relatively common (in 70% of cases), but speech disruptions occur very infrequently (5–7% of the time) (*Ostrowsky et al., 2000*; *Afif et al., 2010*; *Pugnaghi et al., 2011*; *Stephani et al., 2011*; *Mazzola et al., 2017*).

The major impediment in categorizing the role of the insula has been the lack of information regarding the timing of its activation. Its deep location renders it inaccessible to standard non-invasive electrophysiological techniques. We have previously shown that it is possible to obtain direct high spatiotemporal resolution recordings of the right (non-dominant) insula using stereotactically placed depth electrodes, work that allowed us to disambiguate its role in stopping movement relative to activity of the right IFG (*Bartoli et al., 2018*).

Here, we performed direct, invasive recordings of cortical activity from multiple sites across the insula in both hemispheres, in patients undergoing seizure localization for intractable epilepsy, testing the theories generated from the existing literature, namely that the insula acts as a pre-articulatory preparatory region. Given that the insula is thought to be engaged by complex articulation, we used two speech articulation tasks of varying levels of complexity, either reading complex multisyllabic words or naming objects with mono or multisyllabic names. To help disentangle speech-specific neural activity from more general processes, recordings were also performed during listening to speech and during orofacial movements. Taken together, these tasks allow us to map insular regions specifically involved in the preparation and production of speech.

## Results

Participants read visually presented words aloud (n = 27), performed an object naming task (n = 23) of common objects presented as line drawings, listened to speech stimuli as a part of a naming to description task (n = 21) and performed an orofacial praxis task (n = 8), where they silently performed non-speech mouth movements (*Figure 1A*).

### Behavioral performance

Task accuracy for the word reading task was 96.4 ± 5.3% (mean ± SD) with an average response time (RT) of 978 ± 221 ms from word presentation. The RT for the object naming task was 1192 ± 245 ms. 75.1 ± 10.6% of responses in the naming task had both correct articulation and the expected, most common word choice; only these were used for subsequent analysis.

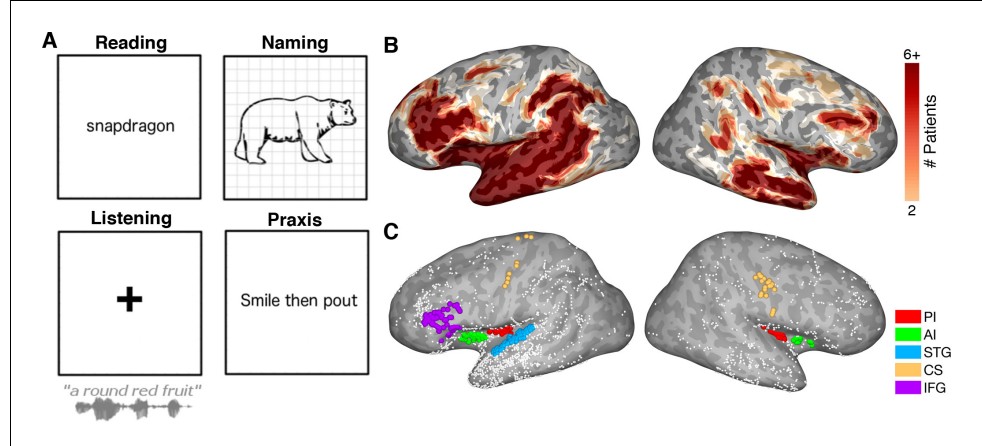

**Figure 1.** Experimental design. (A) Schematic representation of the four tasks. (B) Representative spatial coverage map on a standard N27 inflated surface illustrating how many patients had electrodes in each brain region. (C) Individual electrodes shown on the same brain surface. Colored electrodes represent those included in each ROI for the grouped electrode analyses. PI: Posterior Insula, AI: Anterior Insula, STG: Superior Temporal Gyrus, CS: Central Sulcus, IFG: Inferior Frontal Gyrus.

## Electrode coverage

A plot of the coverage across our standardized brain, accumulating electrode recording zones from each patient (*Figure 1B*) (*Kadipasaoglu et al., 2014*), confirmed good coverage bilaterally in all sub-regions of the insula and across our extra-insular ROIs.

## Chronology of insula activation

To compare the timing of activation of other functional regions with the insula, we used ROIs based on known anatomico-functional parcellation of the insula, separating the short gyri (anterior) and long gyri (posterior) (*Naidich et al., 2004*), and targeted adjacent regions well-established to be involved in speech production, as detailed in the methods: left superior temporal gyrus (STG; primary and secondary auditory cortex), bilateral central sulcus (CS) and left inferior frontal gyrus (IFG) (*Figure 1C*). During both reading and naming, activity in these ROIs was as expected (*Figure 2*; *Figure 2—source data 1*). IFG activation began ~750 ms before speech onset, prior to CS activation. CS activity was maximal at speech onset and, shortly after the onset speech, STG became active.

The posterior insula was active exclusively after speech onset implying that it did not play a role in speech planning. There were no differences in the amplitude of activation with varying levels of articulation complexity - simple monosyllabic names vs. complex read words (Wilcoxon rank sign, 200–600 ms; p=0.083) and multisyllabic naming responses (p=0.898) (*Figure 2—figure supplement 1*). The only difference observed was a duration effect, with longer articulation times and therefore longer activation duration for multisyllabic words (600–1000 ms; p=0.024).

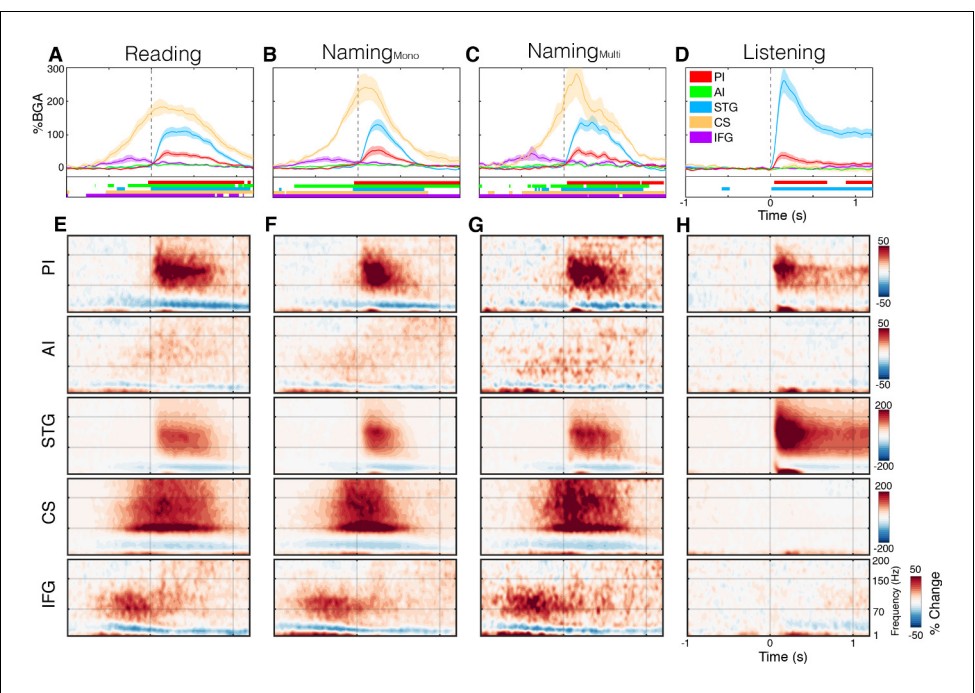

**Figure 2.** Spectrotemporal representations of activity in the ROIs. Broadband gamma activity (**A–D**) and spectrogram (**E–H**) plots of activity within each ROI, averaged across subjects during the complex reading (A,E; n = 27), monosyllabic naming (B,F; n = 23), multisyllabic naming (C,G; n = 23) and listening (D,H; n = 21) tasks. Colored bars under the BGA plots represent regions of significant activation (q < 0.05). Responses are time locked to speech onset in the reading and naming tasks and to the stimulus onset in listening.

The online version of this article includes the following source data and figure supplement(s) for figure 2:

**Source data 1.** Source data for *Figure 2A–D*.
**Figure supplement 1.** Time windowed analysis of activity in the ROIs.
**Figure supplement 2.** Spectrotemporal representations of activity in the praxis task.
**Figure supplement 3.** Evoked oscillations during speech articulation.

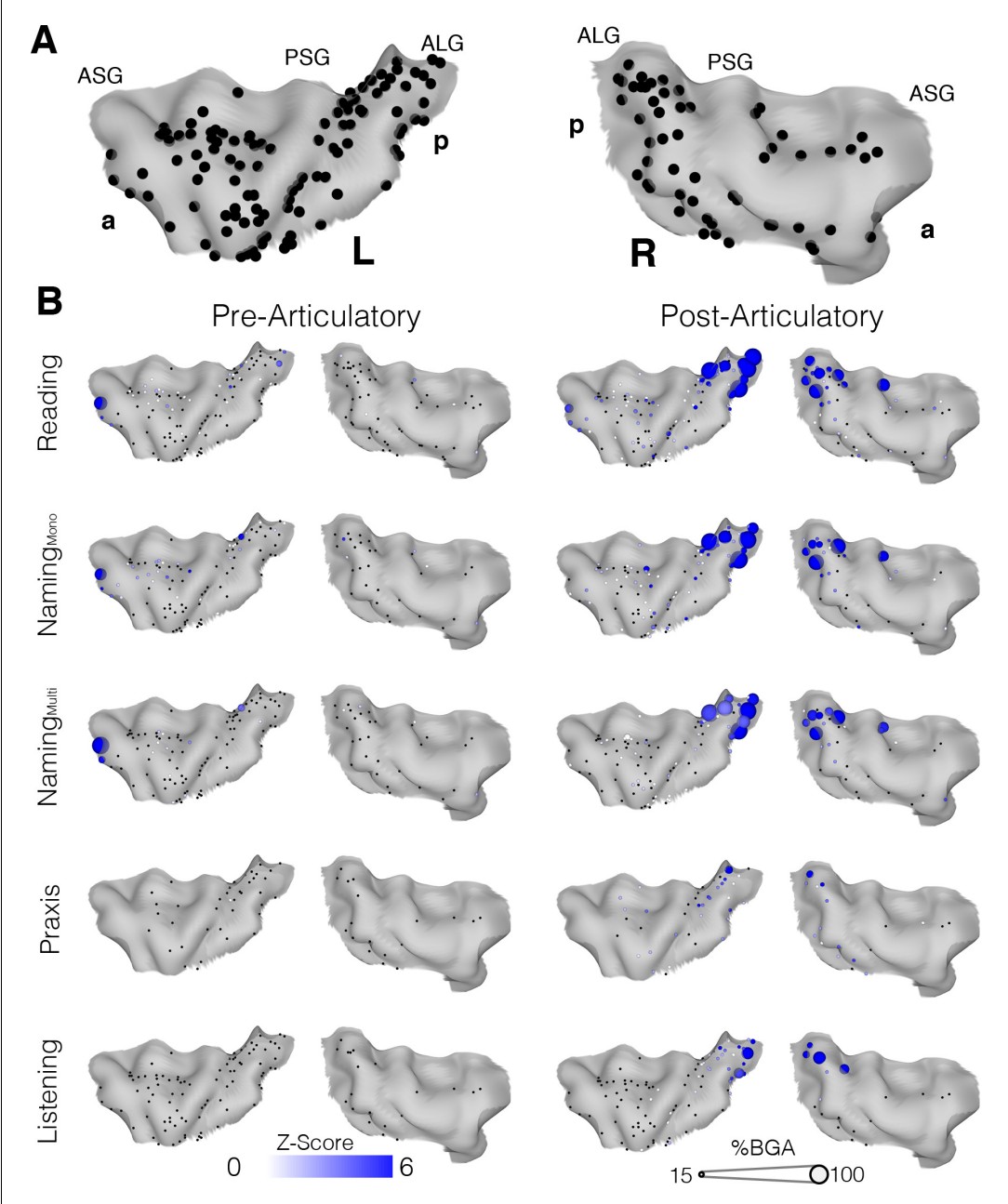

**Figure 3.** Topographic maps of a standardized insula. (**A**) Insula coverage map showing all insula electrodes from patients who did the reading task on a standard N27 pial surface insula. (**B**) Activity maps showing activation above baseline for each task in either the pre (−500 to −100 ms) or post (200 to 600 ms) articulatory time window. ASG: Anterior Short Gyrus, PSG: Posterior Short Gyrus, ALG: Anterior Long Gyrus. Electrodes with a non-significant activation (q > 0.05) shown in black.

The online version of this article includes the following source data for figure 3:

**Source data 1.** Source data for *Figure 3B*.

In posterior insula, the timing of responses resembled those of STG very closely, with similar onset, offset and peak activity times. As expected, from other studies of auditory cortex, the STG responded more strongly to external speech than to self-generated speech (p=0.002) (*Figure 2— figure supplement 1*). The posterior insula, however, showed a significantly greater response during speech production than during speech listening (p=0.019), thus these two adjacent regions are

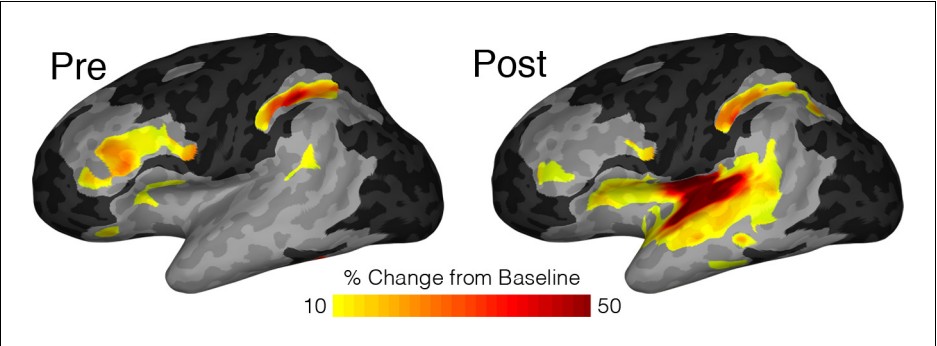

**Figure 4.** MEMA map showing left hemispheric activity in the pre (−500 to −100 ms) and post (200 to 600 ms) articulatory periods during the reading task. Regions are shown for clusters with significant activity (p<0.01, corrected), absolute BGA change of >10% and coverage of at least three patients. Regions excluded for lack of patient coverage are shown in black.

functional dissociable. Further, in the praxis task, we observed significant post-articulatory activation in posterior insula where none was seen in STG (*Figure 2—figure supplement 2*).

The anterior insula showed a very weak though significant activation in both speech articulation tasks, starting shortly after the IFG. This low amplitude response first became significant around 150 ms before speech onset and remained active for the duration of speech, concurrent with the posterior insula and STG. This small but reliable response could represent a low magnitude local processing, but could also represent activity from an active adjacent region, such as the frontal operculum which overlies the insula. Additionally, when considering the evoked activity from anterior insula, we observed no significant response (*Figure 2—figure supplement 3*).

## Topography of insula activation

To evaluate activity in the insula without imposing regional homogenization intrinsic to grouped electrode analysis, and form an accurate spatiotemporal activity map, we also represented the activity of individual insular electrodes on a standardized brain surface (*Figure 3A*). During both reading and naming the magnitude and significance of the activation at individual electrodes were highly consistent. In the post-articulatory period of both the reading and naming tasks, the posterior insula, bilaterally along the anterior long gyri (ALG), showed clusters of electrodes responding with high significance and amplitude, primarily in the superior part of the gyrus (*Figure 3B*; *Figure 3—source data 1*). This region also appeared to be the primary region of activation in the listening task and also showed significant but low amplitude activation during the praxis task.

Overall the anterior insula showed characteristics comparable to those in the grouped analysis, with significant but very low amplitude responses in the post-articulatory period but little to no pre-articulatory activity. The active electrodes seen at the anterior edge of the left insula in the pre-articulatory period were both from one patient and were in close proximity to the frontal operculum. The remaining electrodes with significant responses were very low in amplitude and were scattered across the region, with no clear spatial organization.

To account for variations in sampling density across the insula, we also performed a population level analysis using surface based mixed-effects multilevel analysis (sb-MEMA) (*Conner et al.,*

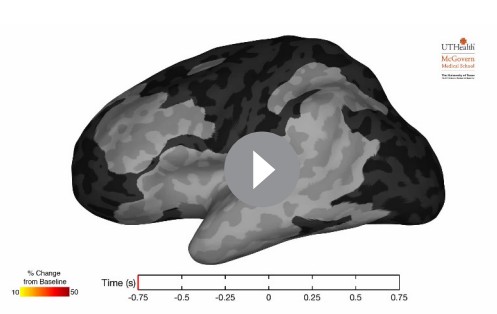

**Video 1.** MEMA video of left hemispheric activity during the reading task. MEMA was run on short, overlapping time windows (150 ms width, 10 ms spacing). Regions are shown for clusters with significant activity (p<0.01, uncorrected), absolute BGA change of >15% and coverage of at least three patients. Regions excluded for lack of patient coverage are shown in black.
https://elifesciences.org/articles/53086#video1

*2011*; *Kadipasaoglu et al., 2014*; *Kadipasaoglu et al., 2015*; *Forseth et al., 2018*) on the reading task (*Figure 4*; *Video 1*). In the pre-articulatory period, we saw prominent activity in the IFG and a cluster across the medial frontal operculum but little to no significant activity anywhere in the insula. By comparison, in the post-articulatory time period, the majority of posterior insula showed substantial activation, accompanied by a large activation across the STG. Additionally, the activation cluster extended across the frontal operculum and spread into part of the superior anterior insula. These results are concordant with our previous analyses in that the posterior insula shows substantial activation during speech production, alongside the superior temporal gyrus. This analysis also showed that while the anterior insula shows little activity the adjacent frontal operculum shows prominent activation.

### Comparing anterior insula and frontal operculum

In fMRI studies of speech and language, the anterior insula is often represented as being active. However, these activation clusters typically encompass both anterior insula and medial frontal operculum (FO). Given that we recorded little activity from electrodes in anterior insula and the MEMA results revealed prominent activity in the FO, it is pertinent to evaluate if FO activity could be the true source of the subtle activation seen in some anterior insular electrodes.

To assess this, we evaluated electrode responses in patients who had electrodes in both the anterior insula and FO. Two examples are shown in *Figure 5*. We noted strong, pre-articulatory BGA responses in FO but low amplitude activity in the anterior insula. FO electrodes showed significant gamma activation that started up to 700 ms before articulation and continued for the duration of articulation (*Figure 5A,B*; *Figure 5—source data 1*).

Due to the oblique trajectories used for sampling the insula, the majority of patients (n = 13) with anterior insula electrodes had a nearby electrode on the same probe (separation 5.7 ± 2.2 mm) that was localized to frontal operculum (*Figure 5C*). The band-limited (70–150 Hz) voltage traces of these electrodes were significantly correlated between the electrode pairs (r = 0.11 ± 0.03, mean ± SE; Wilcoxon rank sign, p=0.008), this correlation was maximal at 0 ms time lag, suggestive of volume conduction between the two regions (*Figure 5—figure supplement 1*). Also, the population level BGA time courses were highly comparable between the two regions (*Figure 5D*).

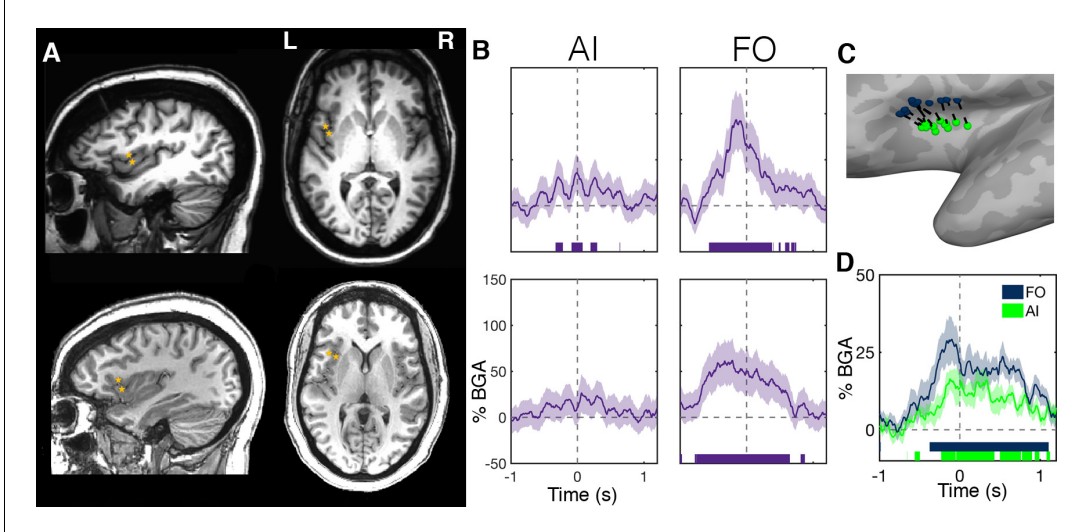

**Figure 5.** Within-patient differences in anterior insula and frontal operculum activity. (**A**) Electrode pairs within two representative patients showing two neighboring electrodes, one within anterior insula (AI) and another within frontal operculum (FO). (**B**) Activity of these electrodes while reading, showing much greater activity within the FO electrodes. Electrode locations (**C**) and BGA (**D**) for neighboring electrode pairs used for within patient comparisons of AI and FO activity. Colored bars under the BGA plots represent regions of significant activation (q < 0.05).

The online version of this article includes the following source data and figure supplement(s) for figure 5:

**Source data 1.** Source data for *Figure 5B*.

**Figure supplement 1.** Time-lagged between-region cross-correlations.

## Posterior insular vs. superior temporal gyrus activity

As we have shown earlier, the posterior insula, particularly the superior ALG, is active in the post-articulatory period during both reading and naming and also during auditory perception and orofacial movements. Posterior insula and STG followed comparable time courses of activation and appeared as one contiguous activation cluster in the MEMA potentially suggesting comparable functional characteristics. However, when we compared the response amplitudes between the reading and listening tasks, within individual electrodes (*Figure 6*; *Figure 6—source data 1*), significantly larger responses were seen along the entire left STG to externally generated over self-generated speech. By contrast, the posterior insula showed greater activation during self-generated speech.

In patients with electrodes in both PI and STG (n = 8), we took the closest electrode pair (separation 11.9 ± 2.9 mm) (*Figure 6B*). In contrast to the AI-FO correlation, band-limited voltage traces in these electrode pairs were not significantly correlated (r = 0.01 ± 0.03, mean ± SE; Wilcoxon rank sign, p=0.74).

## Discussion

Our recordings showed no clear evidence for insular involvement in the preparation for speech. Rather, bilateral posterior insula, particularly the superior anterior long gyri, appear involved in some aspect of monitoring during speech production, a process that is functionally separable from that of the superior temporal gyrus. We find little evidence anterior insula is directly involved in speech or language production, with the neighboring frontal operculum more likely being the true regional activity source.

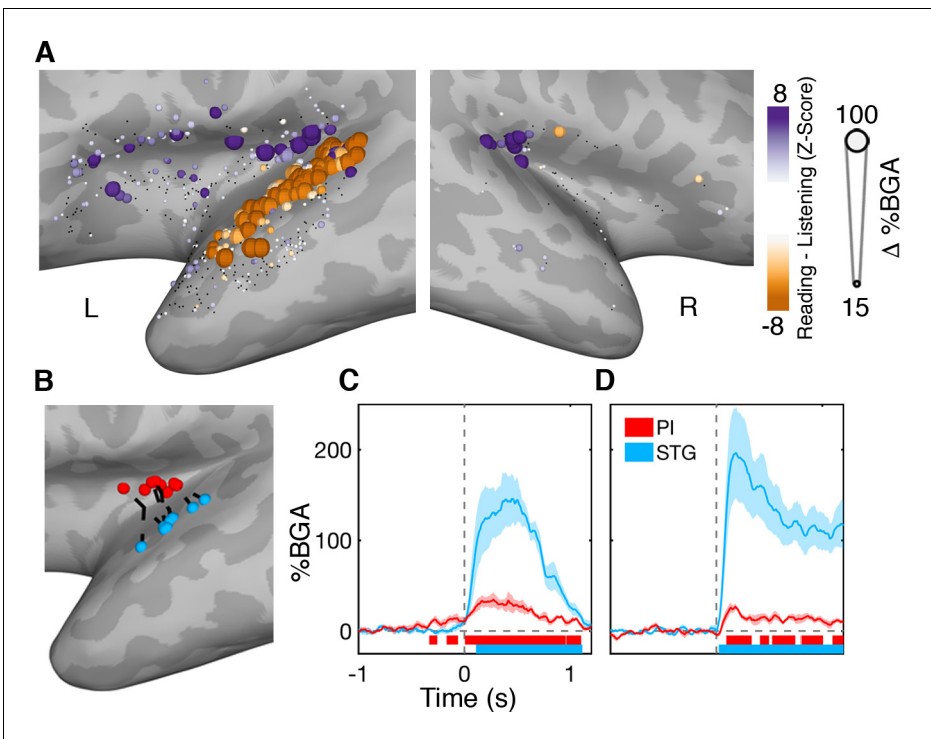

**Figure 6.** Functional dissociation of posterior insula from superior temporal gyrus. (**A**) Contrast map comparing activation during self-generated speech in the reading task to external speech from the listening task in the 200 to 600 ms window. STG showed greater activation during speech perception while PI activity was greater during speech production. Electrodes with a non-significant difference (q > 0.05) shown in black. Electrode locations (**B**) and BGA for reading (**C**) and listening (**D**) for electrode pairs used for within patient comparisons of PI and STG activity. Colored bars under the BGA plots represents regions of significant activation (q < 0.05).

The online version of this article includes the following source data for figure 6:

**Source data 1.** Source data for *Figure 6A*.

## Posterior insula

The posterior insula was strongly active during speech production, with weaker activation during speech perception and much weaker activation during silent mouth movements. The weaker activation during these constituent components of speech compared with speech production could suggest that this region may serve a role in integration of auditory and somatosensory input (*Rodgers et al., 2008*). It could also suggest its involvement in respiratory or laryngeal control (*Ackermann and Riecker, 2010*; *Fedorenko et al., 2015*), aspects we did not control for in these experiments.

This activation profile in posterior insula is the exact opposite of the STG, which is more active during externally produced speech than self-generated speech. It is well known that auditory cortex is suppressed during self-generated speech (*Creutzfeldt et al., 1989*; *Paus et al., 1996*; *Numminen et al., 1999*; *Chan et al., 2014*) and self-generated sounds more generally (*Rummell et al., 2016*; *Singla et al., 2017*), likely as a result of interactions between auditory and non-auditory sensory feedback in auditory cortex. While STG is more active during externally produced speech than self-generated speech, posterior insular activity does not suppress in response to articulation and rather, is more active during self-generated speech.

The notion of that the posterior insula is involved in somatosensory processing is broadly consistent with lesional studies of dysarthric (*Baier et al., 2011*) and dysphagic (*Daniels et al., 1996*) populations and with stimulation studies that result in orofacial sensations (*Pugnaghi et al., 2011*). Our higher spatiotemporal resolution allows us to resolve these properties of the insula better than fMRI studies of mouth movement related activity in this region (*Bonilha et al., 2006*; *Fedorenko et al., 2015*).

Lesional analysis in cases of apraxia of speech, attribute the primary cause to be either (i) disruption of pre-articulatory planning in the superior left posterior short gyrus, (also called the precentral gyrus of the insula), (ii) disruption of pre-articulatory planning in IFG or (iii) impairment of audio-motor integration. Our study provides evidence to rule out the first possibility. Given the lack of pre-articulatory activity shown in this study and the lack of any relationship of activation to the complexity of articulation, it is unlikely that lesions of this region are crucial for AOS. (*Kent, 2000*; *Baldo et al., 2011*). While our results are suggestive of audio-motor integration in the ALG, we do not have direct evidence of this function (*Kent and Rosenbek, 1983*; *Rogers et al., 1996*; *Maas et al., 2015*). Thus, our findings best support AOS representing a disruption of the IFG (*Hillis et al., 2004*; *Fedorenko et al., 2015*).

In summary, the posterior insula (i) lacks pre-articulatory activity, (ii) lacks complexity sensitivity (*Baldo et al., 2011*), (iii) is activated by externally produced sounds and (iv) by non-speech mouth movements. Taken together these findings are suggestive of a sensory monitoring region - congruent with the role of the insula in auditory-somatosensory integration (*Rodgers et al., 2008*) where both auditory and somatosensory activity in rodent insula is maximal during coincident presentation, comparable to what we see during human self-generated speech.

## Anterior insula

The anterior insula is a common area of fMRI activation during speech and language studies. However, these do not show a clear delineation between insula and the operculum in their activation clusters. Our results reveal that the majority of activity lies on the side of the operculum rather than anterior insula. The frontal operculum is also the only peri-insular region in this study with significant and substantial pre-articulatory gamma activity, the timing of which would implicate it as a preparatory region, a role that has traditionally been attributed to the insula. This agrees with stimulation studies of the operculum as disruption of this region has been shown to lead to language disruption (*Mălîia et al., 2018*).

Our electrodes have a center-to-center separation of ~4 mm and this distance between neighboring electrodes allowed us to distinguish the highly active frontal operculum from the minimally active anterior insula. In most modern fMRI, a smoothing kernel of 4–8 mm is used (*Mikl et al., 2008*) which would likely remove the distinction between these two regions. The interpretation of fMRI data derived from peri-insular cortex should therefore consider the degree of smoothing, the use of surface-based smoothing (*Saad and Reynolds, 2012*) and use of individualized patient ROI masks (*Fedorenko et al., 2015*).

In summary, we find that the insula does not serve pre-articulatory preparatory roles, and that bilateral posterior insular cortices may function as auditory and somatosensory integration or monitoring regions. Our findings, analyzed several different ways, have implications for existent models of language production in humans and for the pathophysiology of speech disorders following brain injury.

## Materials and methods

### Participants

Twenty-seven patients (14 male, 18–50 years, 8 left handed) participated in the experiments after written informed consent was obtained. All experimental procedures were reviewed and approved by the Committee for the Protection of Human Subjects (CPHS) of the University of Texas Health Science Center at Houston as Protocol Number HSC-MS-06–0385. Inclusion criteria were that the participants were an English native speaker, had at least one electrode contact localized in the insular long or short gyri and that the insula was not identified as a seizure onset zone.

### Electrode implantation and data recording

Recordings were acquired from stereo-electroencephalographic (sEEG) electrodes (PMT corporation, Chanhassen, Minnesota) implanted for clinical purposes of seizure localization in patients with pharmaco-resistant epilepsy using a Robotic Surgical Assistant (ROSA; Medtech, Montpellier, France) (*Tandon et al., 2019*). sEEG probes were 0.8 mm in diameter, with 8–16 electrode contacts, each of which was a platinum-iridium cylinder, 2.0 mm in length and separated from the adjacent contact by 1.5–2.43 mm. Thus, the center-to-center distance between the electrode contacts was 3.5–4.43 mm. Each patient had multiple (12-16) such probes implanted.

Following implantation, electrodes were localized by co-registration of pre-operative anatomical 3T MRI and post-operative CT scans using a cost function in AFNI (*Cox, 1996*). Electrode positions were projected onto a cortical surface model generated in FreeSurfer (*Dale et al., 1999*), and displayed on the cortical surface model for visualization (*Pieters et al., 2013*). sEEG data were collected using the NeuroPort recording system (Blackrock Microsystems, Salt Lake City, UT) digitized at 2 kHz. They were imported into MATLAB initially referenced to the white matter channel used as a reference by the clinical acquisition system, visually inspected for line noise, artefacts and epileptic activity. Electrodes with excessive line noise or localized to sites of seizure onset were excluded. Each electrode was re-referenced offline to the common average of the remaining channels. Trials contaminated by inter-ictal epileptic spikes were discarded.

### Stimuli and experimental design

Participants read visually presented words aloud (n = 27), performed an object naming task (n = 23) of common objects presented as line drawings, listened to speech stimuli as a part of a naming to description task (n = 21) and performed an orofacial praxis task (n = 8), where they silently performed non-speech mouth movements (*Figure 1A*).

#### Word Reading

Fifty-eight unique words were visually presented in a pseudorandom order with no repetition and patients read the words aloud. Stimuli were presented using Python v2.7, on a 15.4' LCD screen positioned at eye-level, 2–3' from the patient, for 2000 ms with an inter-stimulus interval of 3000 ms. Black, lower-case text (Calibri, height 150 pixels) centered on a 2880 × 1800 pixel white background was used. To create high articulatory complexity and to maximally engage the insula (*Baldo et al., 2011*), all words used had three or more syllables, an initial CCV phoneme structure, and a high articulatory travel (e.g. snapdragon, globalization, claustrophobia).

#### Object Naming

To enable comparisons of a range of articulatory complexities and allow for word selection processes, participants were presented with visual stimuli selected from a standardized set of line drawings (*Snodgrass and Vanderwart, 1980*; *Kaplan et al., 1983*) and instructed to verbally name the objects (*Conner et al., 2014*; *Forseth et al., 2018*). Stimuli were presented in two recording

sessions, each containing presentation of 165 unique images, in a pseudorandom order, that were either coherent images or their spatially scrambled versions. Stimuli were presented using Python v2.7 at a size of 1000 × 1000 pixels centered on a 2880 × 1800 pixel white background on a 15.4' LCD screen positioned at eye-level, 2–3' from the patient, for 1500 ms with an inter-stimulus interval of 3000 ms. Unlike the reading task, this task did not require a single specific response, allowing the patient to choose the response to any given stimulus (e.g. Pelican vs. Bird, Rhinoceros vs. Rhino). Only trials with the most commonly produced word for each stimulus were used for analysis. If all responses were given using expected answers this resulted in 112 monosyllabic responses and 37 multisyllabic (3+ syllable) responses.

### Listening to speech
Participants listened to recorded phrases that described common objects (*Hamberger and Seidel, 2003*; *Forseth et al., 2018*). A total of 72 unique speech stimuli were presented in a pseudorandom order, balanced with an equal number of trials with male and female speakers. Auditory stimuli were played using stereo speakers (44.1 kHz, 15' MacBook Pro 2008) with an inter-stimulus interval of 5000 ms.

### Orofacial Praxis
Participants were cued to perform various orofacial movements silently (e.g. smile then pout, stick tongue out straight) based on the stimuli from *Fedorenko et al. (2015)*. There were 12 unique movements, repeated five times each in a pseudorandom order resulting in a total of 60 trials. Instructions were presented using the same style visual stimuli and timings as in the reading task.

## Audio and video analysis
Continuous audio recordings were carried out using an omnidirectional microphone (30–20,000 Hz response, 73 dB SNR, Audio Technica U841A) placed adjacent to the presentation laptop. These recordings were analyzed offline to transcribe patient responses and manually select the onset of audible speech. In participants who performed the praxis task, we performed video recordings of both reading and praxis tasks to obtain timing of articulation onset and to allow a comparison between the timing of the onset of articulation and that of audible speech. This comparison showed that in the reading task, articulation onset preceded audible speech by 108 ± 127 ms.

## Signal analysis
A total of 5312 electrode contacts were implanted in these patients, 1807 of these were excluded from analysis due to proximity to the seizure onset zone, excessive inter-ictal spikes or line noise. Insular electrodes were selected manually, based on anatomical criteria, after localization of the electrodes' CT artifacts relative to a pre-operative MRI scan. Electrodes in either the anterior (short) gyri or posterior (long) gyri of the insula, as separated by the central sulcus (*Naidich et al., 2004*), were selected for analysis. 52 electrodes in 20 patients were localized to anterior insula (LH n = 40 (16), RH n = 12 (5); # electrodes (# patients)) and 53 electrodes in 14 patients were localized to posterior insula (LH n = 30 (9), RH n = 23 (7)). For the remainder, electrodes were indexed to the closest node on a standardized cortical surface in patient-space to enable grouped representation and analysis (*Saad and Reynolds, 2012*). Regions of interest (ROIs) derived using cortical parcellation from the Human Connectome Project parcellation (*Glasser et al., 2016*) on the standardized surface were used to select electrodes for further analysis. These were the (i) superior temporal gyrus (STG) ROI comprised of: A1, PBelt, MBelt and LBelt in the left hemisphere (n = 106 (16)); (ii) the central sulcus (CS) ROI comprised of areas 3a, 3b and 4 bilaterally (n = 31 (5)). (iii) the inferior frontal gyrus (IFG) ROI comprised of the areas 45, IFSp and IFSa in the left hemisphere (n = 70 (17)). A post-hoc medial frontal operculum ROI (n = 13 (13)) was defined using the same techniques as the insular ROIs, manually selecting electrodes from individual patient MRIs based on anatomical criteria (*Naidich et al., 2004*).

Analyses were performed by first bandpass filtering raw data of each electrode into broadband gamma activity (BGA; 70–150 Hz) following removal of line noise and its harmonics (zero-phase second-order Butterworth band-stop filters). A frequency domain bandpass Hilbert transform (paired sigmoid flanks with half-width 1.5 Hz) was applied and the analytic amplitude was smoothed

(Savitzky-Golay FIR, third-order, frame length of 151 ms; Matlab 2017a, Mathworks, Natick, MA). BGA was defined as percentage change from baseline level; 500 to 100 ms before the presentation of the visual stimulus in each speech production task and 500 to 100 ms before auditory stimulus presentation for the listening task. Periods of significant activation were tested using a one-tailed t-test at each time point and were accepted at a Benjamini-Hochberg false detection rate (FDR) corrected threshold of q < 0.05. Responses were time aligned to the onset of audible speech production. For the grouped analysis, all electrodes were averaged within each subject and then the between subject averages were used. This minimized the influence of outliers in the grouped data.

To evaluate individual insular electrodes, data were tested by calculating the Z-score of the time period of interest against the baseline period. For articulatory tasks, the time periods used were −500 to −100 ms before speech onset and 200 to 600 ms after speech onset. The listening task was tested from 200 to 600 ms after stimulus onset. Statistical significance was accepted at an FDR corrected threshold of q < 0.05.

To provide statistically robust and topologically precise estimates of BGA, population-level representations were created using surface-based mixed-effects multilevel analysis (sb-MEMA) (*Fischl et al., 1999*; *Conner et al., 2011*; *Kadipasaoglu et al., 2014*; *Kadipasaoglu et al., 2015*; *Forseth et al., 2018*). This method accounts for sparse sampling, outlier inferences, as well as intra- and inter-subject variability to produce population maps of cortical activity. Significance levels were computed at a corrected alpha-level of 0.01 using family-wise error rate corrections for multiple comparisons. The minimum criterion for the family-wise error rate was determined by white-noise clustering analysis (Monte Carlo simulations, 5000 iterations) of data with the same dimension and smoothness as that analyzed (*Kadipasaoglu et al., 2014*). Subsequently, a geodesic Gaussian smoothing filter (3 mm full-width at half-maximum) was applied. Results were further restricted to regions with at least three patients contributing to coverage and BGA percent change exceeding 10%. To produce an activation movie, sb-MEMA was run on short, overlapping time windows (150 ms width, 10 ms spacing) to generate the frames of a movie portraying cortical activity.

To generate event-related potentials (ERPs; *Figure 2—figure supplement 3*), the raw data were band pass filtered (0.1–50 Hz). Speech aligned trials were averaged together and the resultant waveform was smoothed (Savitzky-Golay FIR, third-order, frame length of 151 ms). Periods of significant activity were determined as described previously. All electrodes were averaged within each subject, within ROI, and then the between subject averages were used.

## Acknowledgements

We thank Vitoria Piai and Eleonora Bartoli for assistance with stimulus design and for input into earlier versions of the manuscript. We express our gratitude to all the patients who participated in this study; the neurologists at the Texas Comprehensive Epilepsy Program who participated in the care of these patients; and the nurses and technicians in the Epilepsy Monitoring Unit at Memorial Hermann Hospital who helped make this research possible. This work was supported by the National Institute for Deafness and other Communication Disorders DC014589 and National Institute of Neurological Disorders and Stroke NS098981.

## Additional information

### Funding

| Funder | Grant reference number | Author |
| --- | --- | --- |
| National Institute on Deafness and Other Communication Disorders | DC014589 | Oscar Woolnough Kiefer James Forseth Patrick Sarahan Rollo Nitin Tandon |
| National Institute of Neurological Disorders and Stroke | NS098981 | Oscar Woolnough Kiefer James Forseth Patrick Sarahan Rollo Nitin Tandon |

The funders had no role in study design, data collection and interpretation, or the decision to submit the work for publication.

### Author contributions
Oscar Woolnough, Conceptualization, Data curation, Software, Formal analysis, Investigation, Visualization, Methodology; Kiefer James Forseth, Data curation, Software, Investigation; Patrick Sarahan Rollo, Data curation, Investigation; Nitin Tandon, Conceptualization, Supervision, Methodology, Project administration

### Author ORCIDs
Oscar Woolnough (ID) https://orcid.org/0000-0002-5878-6865
Kiefer James Forseth (ID) http://orcid.org/0000-0003-1624-8329
Nitin Tandon (ID) https://orcid.org/0000-0002-2752-2365

### Ethics
Human subjects: Patients participated in the experiments after written informed consent was obtained. All experimental procedures were reviewed and approved by the Committee for the Protection of Human Subjects (CPHS) of the University of Texas Health Science Center at Houston as Protocol Number: HSC-MS-06-0385.

### Decision letter and Author response
Decision letter https://doi.org/10.7554/eLife.53086.sa1
Author response https://doi.org/10.7554/eLife.53086.sa2

## Additional files

### Supplementary files
• Source code 1. MATLAB plotting function for visualization of source data files.

• Transparent reporting form

### Data availability
Source data files and a MATLAB plotting function have been provided for Figures 2, 3, 5 and 6.

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
