## [Decision Letter]

**Acceptance summary:**

Woolnough et al., use intracranial electrocorticography in 27 patients with intractable epilepsy to describe the role of the insula in processing speech during listening and speaking. The goal of evaluating the role of the insula in speech production and planning is well motivated by past lesion and fMRI studies showing its importance in, for example, apraxia of speech. The study finds that responses in the posterior insula are functionally separable from the superior temporal gyrus (STG). Specifically, insula is more responsive during speech production than during perception, whereas STG shows the opposite effect. Moreover, the posterior insula did not show significant activity prior to speech onset, suggesting that it is not part of the pre-articulatory planning network. The anterior insula showed only very low amplitude activity (before speech onset), which was correlated with activity recorded from the frontal operculum (FO), suggesting that FO may be the true source of this planning activity. Gathering human intracranial recordings of this sort is challenging, yet there are no other methods available that provide a view of speech perception and production with this level of resolution and detail. Using this approach, the paper shows the unexpected result that activity in posterior insula during speaking exceeds that during listening. The current study thus provides us with a rare look at the neural processing underlying speech that has important implications.

**Decision letter after peer review:**

[Editors’ note: a previous version of this study was rejected after peer review, but the authors submitted for reconsideration. The first decision letter after peer review is shown below.]

Thank you for submitting your work entitled "Uncovering the functional anatomy of the insula during speech" for consideration by *eLife*. Your article has been reviewed by 3 peer reviewers, and the evaluation has been overseen by a Reviewing Editor and a Senior Editor.

Our decision has been reached after consultation among the reviewers. Based on these discussions and the individual reviews below, we regret to inform you that your work will not be considered for publication in *eLife* as it stands. However, if you are able to fully address the major concerns, we will consider a resubmission since the reviewers also agree that the work is of substantial potential interest.

This is a clearly written paper that evaluates the role of anterior and posterior insular cortex, as well as overlying inferior frontal gyrus and STG and CS regions, in speech production and perception, using reading naming, listening, and orofacial praxis tasks. Recordings were analyzed from a large number of electrodes, in a large number of patients. The researchers conclude that insular activity is evident only post speech onset, indicating that it does not participate in speech planning.

Some critical limitations led the reviewers to question whether the conclusion is really justifiable. First of all, the main conclusion, that insula does not participate in motor planning, stems from a null result, which is always tricky to interpret. (Is this a sensitivity issue? Or perhaps looking in an inappropriate time window?) Second, the imputation of anterior insular activity to FO is qualitative, as are inferences about functional specialization (e.g., anterior vs posterior insula). A more rigorous statistical treatment would help to show that different regions participate in the tasks in different ways.

Another issue is a lack of detail around the acoustic stimuli, and the possible degree to which acoustic (durational?) differences may influence results. Related to this, the time windows analyzed may have a large impact on what is seen. Reviewer #1 mentions the possibility of low-amplitude early activity that may contain important speech-relevant information, and reviewer #2 indicates that the red curves in Figure 2A-D provide a specific example of how the choice of time window changes interpretation – that production responses last longer that perception responses but this could be due to differences in the stimuli between the tasks.

Finally, as reviewer #3 notes, perhaps data from the praxis task could be leveraged more thoroughly to get more information about the timing of speech motor responses, and to inform analysis of data.

Reviewer #1:

This is a well written paper that evaluates the role of anterior and posterior insular cortex, as well as overlying inferior frontal gyrus and STG and CS regions, in speech production and perception, using reading naming, listening, and orofacial praxis tasks. Recordings were analyzed from a large number of electrodes, in a large number of patients. The results appear to demonstrate insular activity only post speech onset, indicating that it does not participate in speech planning.

I have one major concern with this paper as it stands.

First, the time scales that are discussed are exclusively long ones – 200 ms or more. Other iEEG studies (e.g. Brugge et al., 2003, and others from Matt Howard's group in Iowa) demonstrate early evoked auditory activity, within a few tens of ms of stimulation, in addition to the much later population responses discussed here (and evident on whole-brain noninvasive EEG). The amplitude is much lower than the population response that happens later, but the earlier activity is consistent with synaptic conduction delays. The problem is that these earlier, smaller signals might carry a lot of information, and be functionally important. In other words, there may be planning activity at a lower amplitude in insular cortex, early, but it is being missed.

At a minimum this issue of first evoked response (early) vs population activity (late) needs to be discussed, I think.

Reviewer #2:

Woolnough and colleagues present data on the role of the insula in speech. Using multiple speech and non-speech tasks, they report that high-frequency activity in bilateral posterior insula starts after speech onset during both speaking and listening, which they argue may reflect a monitoring process. In contrast, activation in anterior insula is very low amplitude, starting before speech onset. They suggest that this activity may actually reflect the frontal operculum. In comparison to other cortical speech regions, they conclude that posterior insula is involved in monitoring or sensorimotor processing particularly during speech production.

This is a unique data set, combining intracranial recordings from a relatively large cohort of epilepsy patients. The broad question about the role of the insula in speech is extremely important, as they point out that this area is consistently activated across many studies, yet we do not understand its functions. In general, the analyses are statistically robust, and the use of multiple tasks to compare across speech, articulatory, and sensory processing is important.

However, there are several serious problems with the manuscript. I hope that the following comments are helpful to the authors in their revisions of the paper.

Essential revisions:

1) Most of the major claims of the paper are highly speculative. To be clear, the data themselves seem solid, and for the most part, I think it is possible to make novel claims from the analyses that were done. The problem is that there are massive logical leaps required to get from some of the data, e.g.:

"the activity in posterior insula was exclusively post-speech onset".… "There were no differences in amplitude of activation with varying levels of articulation complexity, comparing the simple monosyllabic words from the naming task against the complex reading […] and multisyllabic naming responses"

to the interpretation, e.g.:

"implying a role not as a planning region but possibly as a monitoring region"

The claim about posterior insula as a monitoring region (or is it auditory and somatosensory integration [subsection “Anterior Insula”]?) has no clear support in the data or analyses. It relies on the fact that response differences from baseline do not begin until after acoustic speech onset, and that articulatory complexity (varying across tasks) does not affect response amplitude. I do not deny that these interpretations could be accurate, but I do not see any analyses that explicitly test them in a falsifiable way.

Here's another major example: the claim that any supra-threshold activation in anterior insula actually reflects neural activity in the frontal operculum is based entirely on a qualitative analysis of electrodes in both regions (Figure 5, subsection “Comparing Anterior Insula and Frontal Operculum”). It seems that the authors want to claim that the low amplitude and proximity to FO means that these responses actually come from FO (subsection “Comparing Anterior Insula and Frontal Operculum”). Yet the analysis in Figure 5 seems to actually suggest that the activity between these regions is completely different. One way to actually evaluate this hypothesis is to simply (cross-)correlate the activity between the regions (can you predict FO from AI, and vice versa?), yet no similar analysis was done here.

Overall, the analyses that were done provide some relatively clear results regarding the localization and timing of activity in the (at least posterior) insula during various speech, motor, and sensory tasks. But in my opinion, the interpretations are grossly overextended. In general, this paper seems to be a series of descriptions of activity, many of which are novel. But it's not clear to me how this changes our understanding of either speech production or speech perception networks in the brain, and specifically toward the stated goal of the paper, which is understanding the role of the insula in these networks.

2) In general, the specific hypotheses are hard to follow. For instance, the hypothesis laid out in the Introduction seems to suggest that the FO/AI link shown later in the paper is an a priori hypothesis. Yet the way the Results section is written, it's not clear if the authors truly suspected this was the case, or whether they observed the low amplitude AI activity and then tried to test whether it actually reflects FO. A related example is that the ROIs briefly mentioned in subsection “Comparing Anterior Insula and Frontal Operculum” (and detailed more in the Materials and methods section) do not state that there was a particular reason (possibly other than a briefly mentioned set of fMRI results) to parcellate the insula into anterior/posterior. Thus, I am left confused about both the hypothesis and how it relates to the previous fMRI and lesion literature mentioned in the Introduction.

Another example: the fourth paragraph of subsection “Comparing Anterior Insula and Frontal Operculum”. The paragraph starts by saying the goal is to disambiguate something (what is not clear – is it STG vs insula, or auditory vs sensorimotor?). At the end of the paragraph, a claim is made that STG and posterior insula are functionally separable. The relative amplitude of activation between speaking and listening in both regions does not really disambiguate the role of these regions, it simply shows that one has higher activity than the other in each task (also without noting the extensive literature on suppressed responses during self-produced speech in areas like STG). The analysis in Figure 6 quantifies this in a slightly different way, though ultimately it is the same analysis, and therefore I do not think strengthens the claim.

Similarly, the interpretation of the FO/AI relationship in the Discussion also presents claims that the present manuscript explains lesion data, but without much clarity as to how it rules out the supposed alternatives (subsection “Posterior Insula”; what makes the alternatives "less likely"?).

In general, an approach like linear mixed-effects modeling would greatly improve the ability to understand the functions of these electrodes/regions, rather than just comparing average amplitudes across tasks that may not be perfectly controlled for features that these neurons care about.

3) It is unclear to me why the authors put so much focus on the FO/AI relationship while not testing the same hypothesis for STG/PI. As far as I am aware, these regions have similarly proximate locations, and depending on the trajectories of the depth electrodes, could theoretically pick up the same signals. Here again, the differences in response amplitude between speaking and listening tasks would not truly disambiguate these regions either in terms of shared signals or functional properties. I believe that additional approaches are necessary for making the FO/AI claim (and perhaps also testing the STG/PI claim), including modeling of the BGA spatial spread and/or single pulse stimulation to look for functionally connected electrodes.

4) Why do Figure 4 and Figure 6 only show data from the left hemisphere? Figure 1 and Figure 3 show clear bilateral coverage, and at times, the authors claim that effects are indeed bilateral. However, the summary paragraph at the end of the Discussion section suddenly claims that these effects are left-lateralized.

Reviewer #3:

In "Uncovering the functional anatomy of the insula during speech" Woolnough et al. employ intracranial electrocorticography in a large cohort of intractable epilepsy patients during listening and speaking tasks to describe the role of the insula in speech processing. They find that, despite previous assertions on the role of the insula in speech production, the insula is not involved in pre-articulatory preparation, but is more likely an auditory and somatosensory integration area that is particularly responsive during self-produced speech compared to externally generated speech. I found this paper to be well-written and exciting, since I have not yet seen any ECoG papers directly address the timing of speech responses in the insular cortex and compare them to other speech-responsive areas. The analysis was relatively straightforward, comparing reading, naming, listening, and praxis tasks (orofacial movements) while recording from the posterior and anterior insula, superior temporal gyrus, central sulcus, and inferior frontal gyrus. The authors assert that the posterior insula, in particular the anterior long gyrus (ALG), is not active during pre-articulatory time periods (pre-speech), but is active during listening. More specifically, it is more active during reading and naming as compared to listening to pre-recorded sounds, while superior temporal gyrus shows the opposite pattern. The ALG is also active during orofacial movements, but to a lesser extent than when these movements produce an overt sound (i.e. during object naming or reading). Activity in this area is also dissociated from activity in the frontal operculum, which appears to be motor/preparatory in nature. This result provides an important missing link into how self-monitoring during speech production is mediated by separate auditory circuits, and serves to reconcile disparate findings in noninvasive modalities. I only have a few comments to improve the manuscript.

1) Overall, the use of "superior temporal gyrus" to mean the entire auditory core, belt, and parabelt areas is fine, but I found it distracting at first since different gyri within the superior temporal gyrus may have quite different functional roles. For example, much of the activity recorded in the STG ROI appears to be in the transverse temporal gyrus (Heschl's gyrus), which could be parcellated as a separate area. I do not suggest that the authors need to do this, rather, it would be good to state upfront (e.g. in the Results section, or in any case before the methods section) that this ROI was a large region encompassing primary, secondary, and tertiary auditory cortex.

2) The authors note that ~75% of responses in the naming task were correctly articulated and used the most common word choice. If the number of trials allows, the authors could consider comparing a subset of the ~25% error trials (incorrect articulations) to correct trials to determine whether the posterior insula shows a difference in response magnitude or timing when the utterance is incorrect. If posterior insula is indeed critical for self-monitoring and is implicated in dysarthria, this could provide some key insight into how that process occurs.

3) In Figure 2, the large scale of activity in motor areas and the superior temporal gyrus means that the color scale is blown out. I would suggest scaling each row of panels from -max to +max within an ROI, since the main comparison being made with these is across tasks, while the comparison across areas can be done using panels A-D.

4) Related to point (1), in Figure 3, the authors show the difference in activation for reading and listening to isolate contributions from externally generated vs. self-generated speech. It is difficult to tell the difference between STG, Heschl's gyrus, and regions of the insula. I would suggest that the authors draw an ROI over the regions that they're classifying as each anatomical area.

5) During the listening task, the patients hear words that are presumably not the same as the words that they generate in the reading and naming tasks. To what extent might these acoustic differences influence their results? Do the authors expect that responses during a playback condition should show the same result as their listening task? An analysis of the acoustic properties of the reading/naming vs. listening sounds could be helpful here.

6) Although fewer participants completed the praxis task, it would be helpful to see this task as a similar plot to Figure 2, in a supplemental figure. Currently the praxis task is shown only in Figure 3B for two averaged time points, so it is difficult to know whether the time-course of activity is similar to the other tasks.

[Editors’ note: minor issues and corrections have not been included, so there is not an accompanying Author response.]

In looking over your final copy, please consider the following points and make any changes you think merited.

The reported data provide modest support for the idea that primary anterior insula (AI) activity can be explained by a source in FO, and insula is not involved in pre-articulatory motor planning. The data do show that FO has substantially stronger activity, but that is not the same as showing that there is no activity in AI. The authors also present a cross-correlation analysis showing a double dissociation between activity in posterior and anterior insula. Yet the support for the idea that activity in AI is actually coming from FO rests on the observation of what is a low-- albeit significant-- correlation (r=0.11). Phrased another way, FO activity only explains 1.2% of the variance in AI responses. Yet, the amplitude of AI is likely close to the noise floor of the recordings; thus, one may not expect to observe a particularly high correlation coefficient even if FO is the source of AI responses. In the end, this argument is not particularly convincing. The problem is a classic: a negative result is essentially impossible to prove. That said, negative results are important to report and have extraordinary value in combatting publication bias, so should be included in the published paper.

Regardless of how the results are to be interpreted, the authors should include substantially more information about how the cross-correlations were done. Was it pairwise for each electrode in each region? Or for the most proximal electrodes in each region? Or averaged across electrodes in each region? What was the peak lag, which could help us resolve whether AI activity really reflects FO, (since volume conduction should result in a 0 lag)? Providing such details will help readers understand the results and their significance.

---

## [Author Response]

[Editors’ note: the author responses to the first round of peer review follow.]

Reviewer #1:I have one major concern with this paper as it stands.First, the time scales that are discussed are exclusively long ones – 200 ms or more. Other iEEG studies (e.g. Brugge et al., 2003, and others from Matt Howard's group in Iowa) demonstrate early evoked auditory activity, within a few tens of ms of stimulation, in addition to the much later population responses discussed here (and evident on whole-brain noninvasive EEG). The amplitude is much lower than the population response that happens later, but the earlier activity is consistent with synaptic conduction delays. The problem is that these earlier, smaller signals might carry a lot of information, and be functionally important. In other words, there may be planning activity at a lower amplitude in insular cortex, early, but it is being missed.At a minimum this issue of first evoked response (early) vs population activity (late) needs to be discussed, I think.

We performed a new evoked response analysis, at the reviewer’s behest, looking across our ROIs. We find no evidence of significant pre-articulatory potentials in either insular region. We have added text in the methods relevant to the ERP analysis and have incorporated a new Figure 2—figure supplement 3 in the manuscript

“To generate event related potentials (ERPs; Figure 2—figure supplement 3) the raw data were band pass filtered (0.1 – 50 Hz). Speech aligned trials were averaged together and the resultant waveform was smoothed (Savitzky-Golay FIR, third order, frame length of 151 ms). Periods of significant activity were determined as described previously. All electrodes were averaged within each subject, within ROI, and then the between subject averages were used.”

Additionally, to help improve the presentation of our time resolved data we have generated a new 4D representation of cortical activation using MEMA, using smaller time windows (150 ms windows, 10ms center offset) (Video 1), that shows the progression of pre-articulatory activity from IFG to FO but does not result in AI activation.

Reviewer #2:Essential revisions:1) Most of the major claims of the paper are highly speculative. To be clear, the data themselves seem solid, and for the most part, I think it is possible to make novel claims from the analyses that were done. The problem is that there are massive logical leaps required to get from some of the data, e.g.:"the activity in posterior insula was exclusively post-speech onset".… "There were no differences in amplitude of activation with varying levels of articulation complexity, comparing the simple monosyllabic words from the naming task against the complex reading […] and multisyllabic naming responses"to the interpretation, e.g.:"implying a role not as a planning region but possibly as a monitoring region"The claim about posterior insula as a monitoring region (or is it auditory and somatosensory integration [subsection “Anterior Insula”]?) has no clear support in the data or analyses. It relies on the fact that response differences from baseline do not begin until after acoustic speech onset, and that articulatory complexity (varying across tasks) does not affect response amplitude. I do not deny that these interpretations could be accurate, but I do not see any analyses that explicitly test them in a falsifiable way.

Our claims are based on a reasonable extension of the results we have found – however, we concede the reviewers point as the insula may have other functions beyond those assessable by the experimental paradigms presented here. We have therefore eliminated any reference to the role of posterior insula in monitoring in the abstract and in the Results sections. In the discussion, we summarize salient findings and have added this text, so that future efforts can be informed by our perspective.

“In summary, the posterior insula (i) lacks pre-articulatory activity, (ii) lacks complexity sensitivity (Baldo et al., 2011), (iii) is activated by externally produced sounds and (iv) by non-speech mouth movements. Taken together these findings are suggestive of a sensory monitoring region – congruent with the role of the insula in auditory-somatosensory integration (Rodgers et al., 2008) where both auditory and somatosensory activity in rodent insula is maximal during coincident presentation, comparable to what we see during human selfgenerated speech.”

Here's another major example: the claim that any supra-threshold activation in anterior insula actually reflects neural activity in the frontal operculum is based entirely on a qualitative analysis of electrodes in both regions (Figure 5, subsection “Comparing Anterior Insula and Frontal Operculum”). It seems that the authors want to claim that the low amplitude and proximity to FO means that these responses actually come from FO (subsection “Comparing Anterior Insula and Frontal Operculum”). Yet the analysis in Figure 5 seems to actually suggest that the activity between these regions is completely different. One way to actually evaluate this hypothesis is to simply (cross-)correlate the activity between the regions (can you predict FO from AI, and vice versa?), yet no similar analysis was done here.

Motivated by the reviewers’ comments, we performed a cross-correlation analysis of the gamma band limited voltage trace data from electrode pairs within these adjacent regions (subsection “Comparing Anterior Insula and Frontal Operculum”, subsection “Posterior Insular vs. Superior Temporal Gyrus activity”). As anticipated, we found significant correlation between the signals in AI and FO (r = 0.11, p = 0.008). In contrast, and also as per expectations, in the PI-STG analysis, we found no significant correlation (r = 0.01, p = 0.74). This text has been added to the Results section and Figure 5 and Figure 6 have now been modified as below.

“Due to the oblique trajectories used for sampling the insula, a majority of patients (n=13) with anterior insular electrodes also had an electrode on the same probe (separation 5.7 ± 2.2 mm) that was localized to frontal operculum (Figure 5C). The band-limited (70-150Hz) voltage traces at these electrodes were significantly correlated between electrode pairs (r = 0.11 ± 0.03, mean ± SE; Wilcoxon rank sign, p = 0.008). Also, the population level BGA time courses were highly comparable between the two regions (Figure 5D).

In patients with electrodes in both PI and STG (n=8) we correlated activity in the closest electrode pair (separation 11.9 ± 2.9 mm) (Figure 6B). In contrast to the AI-FO correlation, band-limited voltage traces in these electrode pairs were not significantly correlated (r = 0.01 ± 0.03, mean ± SE; Wilcoxon rank sign, p = 0.74).”

Overall, the analyses that were done provide some relatively clear results regarding the localization and timing of activity in the (at least posterior) insula during various speech, motor, and sensory tasks. But in my opinion, the interpretations are grossly overextended. In general, this paper seems to be a series of descriptions of activity, many of which are novel. But it's not clear to me how this changes our understanding of either speech production or speech perception networks in the brain, and specifically toward the stated goal of the paper, which is understanding the role of the insula in these networks.

This study was initially motivated by the prevalent literature that associates speech production and insula, derived from results of lesional (Dronkers, 1996; Marien et al., 2001; Ogar et al., 2006; Itabashi et al., 2016) and functional imaging (Mutschler et al., 2009; Adank, 2012; McGettigan et al., 2013; Ardila et al., 2014; Oh et al., 2014) studies. The Dronkers 1996 paper linking the anterior insula to disruption of speech has now amassed >1300 citations and anterior insula has been included in several high-profile models of speech and language production (Hickok and Poeppel, 2007 (cited 3474 times) and Dehaene, 2009). Further, the role of the insula as a pre-articulatory node has been used as an explanation of apraxia of speech (Ogar et al., 2006; Baldo et al., 2011); Contrary to the predictions of these models (all derived from studies that use techniques without the temporal resolution to confirm this), we do not find significant pre-articulatory activity that originates in the insula; more specifically, we show that the multitude of fMRI studies that have been interpreted to show that the anterior insula generates activity during speech production have misattributed activity from the frontal operculum to the anterior insula.

2) In general, the specific hypotheses are hard to follow. For instance, the hypothesis laid out in the Introduction seems to suggest that the FO/AI link shown later in the paper is an a priori hypothesis. Yet the way the Results section is written, it's not clear if the authors truly suspected this was the case, or whether they observed the low amplitude AI activity and then tried to test whether it actually reflects FO.

We thought we were clear in our statements in the section the reviewer raises. We have included the following in the introduction to clarify our initial hypothesis.

“Here, we performed direct, invasive recordings of cortical activity from multiple sites across the insula in both hemispheres, in patients undergoing seizure localization for intractable epilepsy, testing the theories generated from the existing literature, namely that the insula acts as a pre-articulatory preparatory region.”

In the Introduction, we refer to the IFG as confounding functional imaging and lesional studies of the SPG. The fMRI literature shows speech related activation clusters which are attributed to anterior insula but we believe originate from FO. Lesional studies (Hillis et al., 2004) implicate IFG as the likely alternative to insula as the cause of apraxia of speech. Therefore, we were also concerned about the possibility of it contaminating the signals recorded in insula using sEEG electrodes. Fortunately, γ band signal is highly focal and falls off rapidly with distance. This spatial resolution allowed us to disambiguate the specific roles of AI and FO in our study (Figure 2, Figure 5; Video 1).

A related example is that the ROIs briefly mentioned in subsection “Comparing Anterior Insula and Frontal Operculum” (and detailed more in the Materials and methods section) do not state that there was a particular reason (possibly other than a briefly mentioned set of fMRI results) to parcellate the insula into anterior/posterior. Thus, I am left confused about both the hypothesis and how it relates to the previous fMRI and lesion literature mentioned in the Introduction.

The insula is defined bygross anatomical boundaries between the insular short gyri (anterior) and long gyri (posterior) (Naidich et al., 2004). This is also supported by the existent literature which also invokes separable functional roles – the anterior insula is presumed the primary locus of speech production related activity in functional imaging studies (Mutschler et al., 2009; Adank, 2012; McGettigan et al., 2013; Ardila et al., 2014; Oh et al., 2014). Posterior insula is however more strongly linked to somatosensory and nociceptive processing (Stephani et al., 2011; Garcia-Larrea, 2012).

“To compare the timing of activation of other functional regions with the insula, we used ROIs based on known anatomico-functional parcellation of the insula, separating the short gyri (anterior) and long gyri (posterior) (Naidich et al., 2004), and targeted adjacent regions well-established to be involved in speech production, as detailed in the methods: left superior temporal gyrus (STG; primary and secondary auditory cortex), bilateral central sulcus (CS) and left inferior frontal gyrus (IFG) (Figure 1C).”

Another example: the fourth paragraph of subsection “Comparing Anterior Insula and Frontal Operculum”. The paragraph starts by saying the goal is to disambiguate something (what is not clear – is it STG vs insula, or auditory vs sensorimotor?). At the end of the paragraph, a claim is made that STG and posterior insula are functionally separable. The relative amplitude of activation between speaking and listening in both regions does not really disambiguate the role of these regions, it simply shows that one has higher activity than the other in each task (also without noting the extensive literature on suppressed responses during self-produced speech in areas like STG). The analysis in Figure 6 quantifies this in a slightly different way, though ultimately it is the same analysis, and therefore I do not think strengthens the claim.

Our goal of this analysis was to show a functional dissociation between PI and STG. While, as has previously been shown, STG is suppressed during self-generated speech the posterior insula is not and in fact shows preferential activation. Our separation of representation between Figure 2 and Figure 6 is to show this effect can be seen at the single electrode level in each ROI and is not an effect of combining responses across the region.

We have rewritten the entire subsection “Chronology of Insular Activation” to improve clarity. We are well aware of the speech induced suppression of auditory cortex and have now included references to this literature on the suppression of self-generated sounds in our Discussion section.

“To compare the timing of activation of other functional regions with the insula, we used ROIs based on known anatomico-functional parcellation of the insula, separating the short gyri (anterior) and long gyri (posterior) (Naidich et al., 2004), and targeting adjacent regions well established to be involved in speech production, as detailed in the methods: left superior temporal gyrus (STG – primary and secondary auditory cortex), bilateral central sulcus (CS) and left inferior frontal gyrus (IFG) (Figure 1C). During both reading and naming, activity in these ROIs was as expected (Figure 2). IFG activation began ~750 ms before speech onset, prior to CS activation. CS activity was maximal at speech onset and shortly after the onset speech, STG became active.

The posterior insula was active exclusively after speech onset implying that it did not play a role in speech planning. There were no differences in the amplitude of activation with varying levels of articulation complexity – simple monosyllabic names vs. complex read words (Wilcoxon rank sign, 200-600ms; p=0.083) and multisyllabic naming responses (p=0.898) (Figure 2—figure supplement 1). The only difference observed was a duration effect, with longer articulation times and therefore longer activation duration for multisyllabic words (600-1000ms; p=0.024). In posterior insula, the timing of responses resembled those of STG very closely, with similar onset, offset and peak activity times. As expected, from other studies of auditory cortex, the STG responded more strongly to external speech rather than to self-generated speech (p=0.002) (Figure 2—figure supplement 1). The posterior insula however showed a significantly greater response during speech production than during speech listening (p=0.019) – thus these two adjacent regions are functional dissociable.

The anterior insula showed a very weak though significant activation in both speech articulation tasks, starting shortly after the IFG. This low amplitude response first became significant around 150 ms before speech onset and remained active for the duration of speech, concurrent with the posterior insula and STG. This small but reliable response could represent a low magnitude local processing, but could also represent activity from an active adjacent region, such as the frontal operculum which overlies the insula.”

Discussion section

“This activation profile in PI is the opposite of STG. It is well known that auditory cortex is suppressed during self-generated speech (Creutzfeldt et al., 1989; Paus et al., 1996; Numminen et al., 1999; Chan et al., 2014) and self-generated sounds (Rummell et al., 2016; Singla et al., 2017), likely as a result of interactions between auditory and non-auditory sensory feedback in auditory cortex. While STG is more active during externally produced speech than self-generated speech, posterior insular activity does not suppress in response to articulation and rather, is more active during self-generated speech.”

Similarly, the interpretation of the FO/AI relationship in the Discussion also presents claims that the present manuscript explains lesion data, but without much clarity as to how it rules out the supposed alternatives (subsection “Posterior Insula”; what makes the alternatives "less likely"?).

That is not exactly what we have said: Lesional analysis in cases of apraxia of speech, attribute the problem to be (i) disruption of pre-articulatory planning in insula, (ii) disruption of pre-articulatory planning in IFG or (iii) impairment of audio-motor integration. Our study provides the first clear evidence to rule out the first possibility and instead invokes IFG or FO disruption. We have made modifications to the relevant paragraph to prevent any misinterpretations.

“Lesional analysis in cases of apraxia of speech, attribute the primary cause to be either (i) disruption of prearticulatory planning in the superior left posterior short gyrus, (also called the precentral gyrus of the insula), (ii) disruption of pre-articulatory planning in IFG or (iii) impairment of audio-motor integration. Our study provides evidence to rule out the first possibility. Given the lack of pre-articulatory activity shown in this study and the lack of any relationship of activation to the complexity of articulation, it is unlikely that lesions of this region are crucial for AOS. (Kent, 2000; Baldo et al., 2011). While our results are suggestive of audio-motor integration in the ALG we do not have direct evidence of this function (Kent and Rosenbek, 1983; Rogers et al., 1996; Maas et al., 2015). Thus, our findings best support AOS representing a disruption of the IFG (Hillis et al., 2004; Fedorenko et al., 2015).”

3) It is unclear to me why the authors put so much focus on the FO/AI relationship while not testing the same hypothesis for STG/PI. As far as I am aware, these regions have similarly proximate locations, and depending on the trajectories of the depth electrodes, could theoretically pick up the same signals. Here again, the differences in response amplitude between speaking and listening tasks would not truly disambiguate these regions either in terms of shared signals or functional properties. I believe that additional approaches are necessary for making the FO/AI claim (and perhaps also testing the STG/PI claim), including modeling of the BGA spatial spread and/or single pulse stimulation to look for functionally connected electrodes.

We were focused on the AI and the FO given the principal role the AI has been attributed in prior explanations of speech apraxia. We have now performed and included a cross-correlation analysis of the voltage trace data from electrode pairs within these abutting regions (subsection “Comparing Anterior Insula and Frontal Operculum”, subsection “Posterior Insular vs. Superior Temporal Gyrus activity”). In the AI-FO comparison there is a significant correlation between the signals and this is not seen in the PI-STG comparison.

“Due to the oblique trajectories used for sampling the insula, the majority of patients (n=13) with anterior insula electrodes had a nearby electrode on the same probe (separation 5.7 ± 2.2 mm) that was localized to frontal operculum (Figure 5C). The band-limited (70-150Hz) voltage traces of these electrodes were significantly correlated between the electrode pairs (r = 0.11 ± 0.03, mean ± SE; Wilcoxon rank sign, p = 0.008). Also, the population level BGA time courses were highly comparable between the two regions (Figure 5D).”

“In patients with electrodes in both PI and STG we took the closest electrode pair (separation 11.9 ± 2.9 mm) (Figure 6B). The band-limited voltage traces in these electrode pairs were not significantly correlated between the electrode pairs (r = 0.01 ± 0.03, mean ± SE; Wilcoxon rank sign, p = 0.74).”

4) Why do Figure 4 and Figure 6 only show data from the left hemisphere? Figure 1 and Figure 3 show clear bilateral coverage, and at times, the authors claim that effects are indeed bilateral. However, the summary paragraph at the end of the Discussion section suddenly claims that these effects are left-lateralized.

We had not included the right hemisphere data given that we did not have sufficient broad coverage of the right hemisphere to run a meaningful MEMA analysis – Specifically, for the MEMA analysis a minimum patient coverage of 3 is generally required in any given region. In deference to the reviewer’s request, we have also represented right hemisphere data in Figure 6 and the ambiguity from the summary paragraph has been corrected.

“In summary, we find that the insula does not serve pre-articulatory preparatory roles, and that bilateral posterior insular cortices may function as auditory and somatosensory integration or monitoring regions.”

Reviewer #3:1) Overall, the use of "superior temporal gyrus" to mean the entire auditory core, belt, and parabelt areas is fine, but I found it distracting at first since different gyri within the superior temporal gyrus may have quite different functional roles. For example, much of the activity recorded in the STG ROI appears to be in the transverse temporal gyrus (Heschl's gyrus), which could be parcellated as a separate area. I do not suggest that the authors need to do this, rather, it would be good to state upfront (e.g. in the Results section, or in any case before the methods section) that this ROI was a large region encompassing primary, secondary, and tertiary auditory cortex.

We have now added this distinction to our Results section:

“left superior temporal gyrus (STG; primary and secondary auditory cortex)”

2) The authors note that ~75% of responses in the naming task were correctly articulated and used the most common word choice. If the number of trials allows, the authors could consider comparing a subset of the ~25% error trials (incorrect articulations) to correct trials to determine whether the posterior insula shows a difference in response magnitude or timing when the utterance is incorrect. If posterior insula is indeed critical for self-monitoring and is implicated in dysarthria, this could provide some key insight into how that process occurs.

As we state in the Materials and methods section, in naming trials patients were not constrained in their answer. To allow us to assure specific articulations were mono or multi-syllabic we only analyzed trials with answers that were of the expected words for a given stimulus. >95% of the responses were correctly articulated but ~25% of these made a word choice that was not the most commonly associated word for the presented visual stimulus (e.g. bird instead of pelican), thus, for reasonable grouping of data and population level analyses, we excluded these trials with variations in responses.

3) In Figure 2, the large scale of activity in motor areas and the superior temporal gyrus means that the color scale is blown out. I would suggest scaling each row of panels from -max to +max within an ROI, since the main comparison being made with these is across tasks, while the comparison across areas can be done using panels A-D.

STG and CS have been rescaled for better visualization.

4) Related to point (1), in Figure 3, the authors show the difference in activation for reading and listening to isolate contributions from externally generated vs. self-generated speech. It is difficult to tell the difference between STG, Heschl's gyrus, and regions of the insula. I would suggest that the authors draw an ROI over the regions that they're classifying as each anatomical area.

In Figure 3 we are purely showing the insula, with STG and Heschl’s gyrus hidden as they would usually be obscuring the insula in this representation.

5) During the listening task, the patients hear words that are presumably not the same as the words that they generate in the reading and naming tasks. To what extent might these acoustic differences influence their results? Do the authors expect that responses during a playback condition should show the same result as their listening task? An analysis of the acoustic properties of the reading/naming vs. listening sounds could be helpful here.

Within both reading and listening tasks we have a phonologically diverse range of stimuli. Reading consisted of 60 unique words and listening contained 72 sentences with varying sentence structure and initial word phonemes. There may be variations in posterior insular activity driven by phonological effects, but this does not detract from our principal findings and is beyond the scope of this study.

6) Although fewer participants completed the praxis task, it would be helpful to see this task as a similar plot to Figure 2, in a supplemental figure. Currently the praxis task is shown only in Figure 3B for two averaged time points, so it is difficult to know whether the time-course of activity is similar to the other tasks.

In deference to the reviewer’s wishes, we performed a subgroup analysis using only those patients who performed the praxis task (n = 8). We depict these results as Figure 2—figure supplement 2.

“Further, in the praxis task we observed significant activation in posterior insula where none was seen in STG (Figure 2—figure supplement 2).”